# Genus *Acrostalagmus*: A Prolific Producer of Natural Products

**DOI:** 10.3390/biom13081191

**Published:** 2023-07-30

**Authors:** Ting Shi, Han Wang, Yan-Jing Li, Yi-Fei Wang, Qun Pan, Bo Wang, Er-Lei Shang

**Affiliations:** 1College of Chemical and Biological Engineering, Shandong University of Science and Technology, Qingdao 266590, China; shiting_jia@126.com (T.S.); h15725209196@163.com (H.W.); 15153232290@163.com (Y.-J.L.); kingsley11f115@163.com (Y.-F.W.); 18663100496@163.com (Q.P.); 2State Key Laboratory of Microbial Technology, Institute of Microbial Technology, Shandong University, Qingdao 266237, China; 3School of Life Sciences, Shandong University, Qingdao 266237, China

**Keywords:** *Acrostalagmus*, marine-derived fungi, secondary metabolites, epipolythiodioxopiperazine, bioactivities

## Abstract

*Acrostalagmus* is known for its ability to produce numerous bioactive natural products, making it valuable in drug development. This review provides information on the sources, distribution, chemical structure types, biosynthesis, and biological activities of the compounds isolated from the genus *Acrostalagmus* in the family *Plectosphaerellaceae* from 1969 to 2022. The results show that 50% of the compounds isolated from *Acrostalagmus* are new natural products, and 82% of the natural products derived from this genus are from the marine *Acrostalagmus*. The compounds isolated from *Acrostalagmus* exhibit diverse structures, with alkaloids being of particular importance, accounting for 56% of the natural products derived from this genus. Furthermore, within the alkaloid class, 61% belong to the epipolythiodioxopiperazine family, highlighting the significance of epipolythiodioxopiperazine as a key characteristic structure within *Acrostalagmus*. Seventy-two percent of natural products derived from *Acrostalagmus* display bioactivities, with 50% of the bioactive compounds exhibiting more significant or comparable activities than their positive controls. Interestingly, 89% of potent active compounds are derived from marine fungi, demonstrating their promising potential for development. These findings underscore *Acrostalagmus*, particularly the marine-derived genus *Acrostalagmusas*, a valuable source of new bioactive secondary metabolites, and emphasize the vast resource importance of the ocean.

## 1. Introduction

*Acrostalagmus* is a genus of ascomycete fungi in the class *Sordariomycetes*, order *Glomerellales*, family *Plectosphaerellaceae*. There are four species (*A. annulatus*, *A.* cf. *luteoalbus*, *A.* cf. *luteoalbus* CK1, *A. luteoalbus*) of the genus *Acrostalagmus* in the National Center for Biotechnology Information (NCBI) database (https://www.ncbi.nlm.nih.gov/data-hub/taxonomy/tree/?taxon=461148 (accessed on 27 July 2023)). The colony of *Acrostalagmus* is brick red, because of its red spores, with white mycelium at edge. The mass production of spores causes the overall colony to present a ring pattern with different shades. With the extension of culture time, the color gradually deepened and darkened, showing rust red [1].

Most of the fungi belonging to the genus *Acrostalagmus* are alkalitolerant [2,3] or alkalophilic [4] fungi, and are widely distributed in different ecological environments, including forest [5], sand ridge state [6], marine [7,8] and polar ecosystems [9]. The genus *Acrostalagmus* can survive in different circumstances due to its ability to produce kinds of enzymes [10,11,12] and secondary metabolites [13] with a variety of bioactivities [6,14,15]. The crude extracts of some *Acrostalagmus* species exhibited significant brine shrimp lethality, as well as antibacterial, antifungal and DPPH radical scavenging activities [9,16], meaning they have potential to produce abundant natural products with remarkable activities. Gas chromatography mass (GC-MS) [17], high-performance liquid chromatography (HPLC)-electrospray ionization (ESI)-MS [18], and ultra-HPLC-MS/MS spectrometry [9] have been used to analyze the secondary metabolites of the genus *Acrostalagmus* and further demonstrate the great ability of this genus to produce bioactive compounds. To date, there has been no summary reviewing the natural products of the *Acrostalagmus* genus. In consideration of the above-mentioned facts, the chemical structure types, sources, distribution, biological activities, and biological synthesis of the compounds isolated from *Acrostalagmus* from 1969 to 2022 are comprehensively reviewed in this paper.

## 2. Terpenoids

The first research on the secondary metabolites isolated from the genus *Acrostalagmus* was performed in 1969 by George A. Ellestad et al. [19]. Two norditerpenes, named LL-Z1271α (**1**) and LL-Z1271γ (**2**) (Figure 1), were isolated from an unidentified *Acrostalagmus* sp. NRRL-3481 [19,20]. In 1971, one norditerpene analogue, LL-Z1271β (**3**), was discovered from the same species by the same research group [21]. In 1974, three other analogs **4**–**6** were obtained from the culture of *Acrostalagmus* NRRL-3481 [22]. Terpenoids **1**–**6** were deduced to be biosynthesized from microbiological degradation of a diterpene, such as labdadienol, through oxidative cleavage between C-12 and C-13 [22] (Figure 2). The absolute configuration of **4** at the location of C-8 was deduced to be 8*R* according to the supposed biosynthesis pathway from compound **6** to **4**. Compound **1** displayed remarkable antifungal activity in vitro against kinds of fungi and in vivo against some experimental ringworm infections in guinea pigs [19]. Additionally, **1** displayed effectiveness against the fungi that cause infection in humans with the minimum inhibitory concentrations (MICs) against *Cryptococcus neoformans* and *Candida albicans* of 2 µg/mL and 8 µg/mL, respectively [23]. Compound **1** was the inhibitor of *Pseudogymnoascus destructans*, which is the fungus that leads to a devastating disease of hibernating bats named white-nose syndrome (WNS), with an MIC value of 15 μg/mL [24]. The cytotoxicity of **1** against the murine P388 lymphocytic leukemia cell line and a series of human cancer cell lines were evaluated and IC_50_ values ranging from 0.14 to 4.1 µg/mL [23] were obtained. Compound **3** also showed cytotoxic activity against human cancer cell line HL-60 with an IC_50_ value of 0.60 µM, with the same level as the positive control epirubicin (IC_50_ = 0.71 µM) [25]. Compound **1**, as a plant growth regulator, showed significant inhibitory activity on the growth of an *Avena coleoptile* section comparable to those of structural analogues, inumakilactones, nagilactones, and podolactones, which showed strong inhibitory activity to the expansion and mitosis of plant cell [26]. At a concentration of 10^−4^ M, Compound **1** significantly inhibited the germination and growth development of three plant species: two monocotyledons (*Allium cepa* and *Hordeum vulgare*) and one dicotyledon (*Lactuca sativa*), with an inhibition rate of over 80%, which is more active than the commercial herbicide LOGRAN^®^, indicating that **1** shows potential as a herbicide template and may serve as a next generation of natural agrochemicals [27]. Compound **1** displayed potent inhibitory activity to the production of IL-1β (interleukin-1β, a proinflammatory cytokine produced primarily by macrophages and monocytes in answer to various stimuli [28]) in the manner of dose-dependent application with an IC_50_ value of 0.049 μM in human whole blood [29,30]. Compound **1** exhibited a much weaker inhibitory effect on leucine uptake than on IL-1β production which suggests that the compound’s action is not a result of general effects on protein synthesis. The inhibition mechanism of **1** is also not because of the ATP-induced release, effects on caspase-1, or a lysosomotrophic effect [29]. Further research on the target for **1** is in progress, which may identify a mechanistically new approach for the treatment of IL-1β associated diseases [29].

## 3. Alkaloids

Melinacidin, a mixture of at least four closely related compounds obtained from the culture broth of the fungus *Acrostalagmus cinnabarinus* var. *melinacidinus*, was first discovered in 1968 and showed antibacterial activity against various of Gram-positive bacteria in vitro [31]. However, melinacidin was ineffective in protecting mice from the infection of *Staphylococcus aureus* when administered subcutaneously at the maximum tolerated dose of 1 mg/mL [31]. The mechanism of antibacterial activity of melinacidin was studied and found to be blocked the synthesis of nicotinic acid and its amide in *Bacillus subtilis* cells. The biosynthetic pathway leading to nicotinic acid was interfered with by melinacidin before the formation of quinolinic acid [32]. The antifungal activity of melinacidin was only exhibited on *nocardia asteroides* and *Blastomyces dermatitidis* with MIC values of 10 and 1000 μg/mL, respectively. Melinacidin displayed inhibition of the growth of KB cells in tissue cultures with an ID_50_ (50% inhibition of protein synthesis) value of 0.014 μg/mL and had marginal in vivo activity in mice infected with Herpes virus [31]. In 1972, melinacidin was separated into three compounds, melinacidins II, III, and IV (**7**–**9**), and their structure characterizations were described [33]. While the certain structures of **7**–**9** were finally determined in 1977 to be epipolythiodioxopiperazines (ETPs) (Figure 3) [34]. Compound **9** showed potent cytotoxicity against murine P388 leukemia cells with an IC_50_ value of 0.05 µM [35]. Compound **9** also exhibited antibacterial activities against methicillin-resistant *S. aureus* (MRSA) and vancomycin-resistant *Enterococcus faecium* (VRE) with the MIC values of 0.7 and 22 μg/mL, respectively. The antibacterial activity of **9** to MRSA exhibited double the activity of the positive control vancomycin (MIC = 1.4 µg/mL) [36].

Chemical investigation of the deep-sea sediment-derived fungus *A. luteoalbus* SCSIO F457 led to the isolation of two new indole diketopiperazines, luteoalbusins A and B (**10** and **11**), as well as eight known diketopiperazines, T988A (**12**), gliocladines C and D (**13** and **14**), chetoseminudins B and C (**15** and **16**) (Figure 3), cyclo(L-Trp-L-Ser) (**35**), cyclo(L-Trp-L-Ala) (**36**), and cyclo(L-Trp-*N*-methyl-L-Ala) (**37**) [37]. The bi-indole diketopiperazines (**10**–**14**) exhibited potent cytotoxicity against four cancer cell lines, SF-268, MCF-7, NCI-H460, and HepG-2, with IC_50_ values ranging from 0.23 to 17.78 µM. The new compounds **10** and **11** showed stronger cytotoxicity against all four tested cancer cell lines than that of the positive control cisplatin [37]. Compounds **10** and **11** also displayed prominent cytotoxic activities against A549, HeLa, and HCT116 cancer cell lines with IC_50_ values ranging from 0.52 to 2.33 µM [42]. Compound **12** was first discovered from a decaying wood derived fungus *Tilachlidium* sp. CANU-T988, and displayed cytotoxicity to P388 leukemia cells with an IC_50_ value of 0.25 µM [43]. Compounds **13** and **14** were first isolated from the submerged wood derived fungus *Gliocladium roseum* 1A and showed nematicidal activities toward *Caenorhabditis elegans* and *Panagrellus redivivus* with ED_50_ (concentrations causing more than 50% mortality after 24 h) values of 200/250 and 200/250 µg/mL, respectively [44]. Compounds **10** and **13** were exhibited antimicrobial activities against *Canidia albicans* and *Aeromonas salmonicida* with MIC values of 12.5/12.5 (**10**) and 25/50 (**13**) µM, respectively [9]. Compounds **15** and **16** were first found from the fungus *Chaetomium seminudum* 72-S-204-1, and **15** showed weak immunosuppressive activity with an IC_50_ value of 24 µg/mL on Con A-induced (T-cells) proliferations of mouse splenic lymphocytes [45]. Compounds **15** and **16** exhibited potent cytotoxic activities against murine lymphoma L5178Y cell line with EC_50_ values of 0.26 and 0.82 µM, respectively, which are more potent than that of the positive control kahalalide F (EC_50_ = 4.3 µM) [46]. Compound **15** also showed obvious enzyme inhibition against mushroom tyrosinase with an IC_50_ value of 31.7 ± 0.2 µM, which is stronger than the inhibitory activity of the positive control kojic acid (IC_50_ = 40.4 ± 0.1 μM) [47].

Two new epipolythiodioxopiperazines (ETPs), chetracins E and F (**17** and **18**), as well as one known congener, chetracin C (**19**), were isolated from the culture extract of *A. luteoalbus* HDN13-530, a fungus obtained from the soil of Liaodong Bay [38]. Compounds **17**–**19** displayed extensive cytotoxic activities toward a series of cancer cell lines A549, HCT116, K562, H1975 and HL-60 with the IC_50_ values ranging from 0.2 to 2.1 µM, and **17** even showed stronger cytotoxicity to H1975 cancer cell line with an IC_50_ value of 0.2 µM than that of positive drug doxorubicin hydrochloride (IC_50_ = 0.8 µM) [38]. One of the reasons **17**–**19** cytotoxicity is possible due their ability to reduce the expressions of Akt, EGFR, and the active forms of Akt, EGFR, Erk, and Stat3 (Hsp90 client oncoproteins) in H1975 cells at the concentration of 0.5 µM, indicating their inhibition to C-terminal Hsp90 [38]. Compound **19** was first isolated from Antarctic soil derived fungus *Oidiodendron truncatum* GW3-13 and showed significant cytotoxicity against a panel of the cancer cell lines HCT-8, Bel-7402, BGC-823, and A2780 with IC_50_ values that ranged 0.49–0.70 µM [48].

Three pairs of new *N*-methoxy-indolediketopiperazines enantiomers, (±)-acrozines A–C (**20**–**25**, Figure 3), were isolated from the marine green alga *Codium fragile* derived endophytic fungus *A. luteoalbus* TK-43 [39]. Four new acrozine-type indolediketopiperazines, acrozines D–G (**26**–**29**, Figure 3), along with six known analogues, pseudellone D (**30**), lasiodipline E (**31**), chetoseminudins B and C (**15** and **16**), T988 C and B (**32** and **33**) (Figure 3), were isolated from the culture extract of the same fungal species TK-43 [40]. Compounds **15**, **16,** and **20**–**33** were evaluated for their antimicrobial activities toward 15 plant pathogenic fungi, one human pathogenic bacterium, and 10 aquatic pathogens. Only (–)-acrozine B (**23**) showed antifungal activity toward the plant pathogen *Fusarium solani* with an MIC value of 32 μg/mL, which is stronger than the activities of its enantiomer **22** and its epimers **20** and **21** (MIC > 64 μg/mL) [39]. These results indicate that the absolute configurations of 3*R*, 6*R* are the key structures to producing antifungal activity. While compound **25** with the same configurations of 3*R*, 6*R* had no antifungal activity, this might suggest the significance of methylene hydroxyl and thiomethyl groups located at C-3 and C-6, respectively, for the antifungal activity. Compounds **30** and **16** showed antibacterial activity against *Edwardsiella icataluri* with MIC values of 3 and 5 μM, respectively, which are comparable to that of the positive control, chloromycetin (MIC = 2 μM) [40]. Compound **32** showed broad-spectrum antibacterial activity and demonstrated more potent activity (MIC = 2 μM) against *Vibrio parphemolyticus* than the positive control chloromycetin (MIC = 12 μM) [40]. The antimicrobial activity of **32** against *Candida albicans* (MIC = 6.25 μM) and *Aeromonas salmonicida* (MIC = 3.125 μM) were comparable to that of positive control ciprofloxacin (MIC = 6.25 μM) [9]. The results indicate that antibacterial activities are significantly reduced (from **30** and **16** to **20**–**29**, **31**, and **15**) when there is a methoxy or methyl substitution at *N*-2. Additionally, antibacterial activity is significantly increased when there is a disulfide bridge (from **33** to **32**) [39,40]. Compound **31** was first discovered from the culture of *Illigera rhodantha* (a flower belongs to Hernandiaceae) derived endophytic fungus *Lasiodiplodia pseudotheobromae* F2, and exhibited strong antibacterial activity toward the clinical strains *Bacteroides vulgates*, *Streptococcus* sp., *Veillonella parvula,* and *Peptostreptococcus* sp., with an MIC value range of 0.12–0.25 μg/mL, comparable or even more significant than that of positive control tinidazole (MIC values range of 0.12–0.5 μg/mL) [49]. T988 A and C (**12** and **32**) showed potent antibacterial activities against *S. aureus*, methicillin-resistant *S. aureus*, and *S. pyogenes* with IC_50_ values of 3.8/5.8, 8.4/5.6, and 1.8/3.1 μM, respectively. It was demonstrated that **12** and **32** exhibited antibacterial synergy in combination with ciprofloxacin, ampicillin, and streptomycin [50]. The biofilm inhibition caused by **12** and **32** in *S. aureus* and *S. pyogenes* was approximately 70% at their MIC and over 60% at one-sixteenth of their MIC, respectively [50]. The mechanism of antibacterial activity in compounds **12** and **32** was explored and it was found that they have the ability to inhibit bacterial transcription/translation in vitro and inhibit the production of staphyloxanthin in *S. aureus* [50].

Compounds **20**–**29** were tested for their anti-acetylcholinesterase (AChE) activity. (±)-Acrozines A had medium anti-AChE activity with IC_50_ value of 9.5 μM, and the chiral split compound, (+)-acrozine A (**20**) (IC_50_ = 2.3 μM) displayed better inhibition than that of (–)-acrozine A (**21**) (IC_50_ = 13.8 μM) and (±)-acrozines A [39]. Compound **26**, which has the same planar structure and different configurations as that of **24** (IC_50_ = 160.6 μM) and **25** (IC_50_ = 121.7 μM), displayed much better AChE inhibitory activity with IC_50_ value of 18.9 μM [39,40]. These bioactivity data showed that compounds with identical planar structure may display different bioactivity and that the selectivity of biological activity is associated with the absolute configuration. Compound **28** showed anti-AChE activity with an IC_50_ value of 8.4 μM [40]. Compound **28** differs from (+)-acrozine B (**22**) (IC_50_ = 78.8 μM) [39] solely in the location of SCH_3_ substitution, indicating the SCH_3_ group at C-3 in **28** is more active than the SCH_3_ group at C-6 in **22**.

The biosynthetic pathway for compounds **7**–**33** is speculated as shown in Figure 4. Diketopiperazines **7**–**33** are biosynthesized through non-ribosomal the peptide synthetase (NRPS) pathway [51], and their biosynthetic precursors might be L-Trp and L-Ala (**7**, **13**, **14**, **17**, **24**–**26** and **29**–**31**), or L-Trp and L-Ser (**7**–**12**, **15**–**23**, **27**, **28**, **32** and **33**) [40,52]. The sulfurs are proposed to be incorporated into the cyclopeptide frameworks (**7**–**19**, **21**–**23** and **28**–**33**) by CYP450 monooxygenase and a specialized glutathione *S*-transferase which is similar to that in gliotoxin (GT) [48,51,53,54], and the intramolecular disulfides are generated by FAD-dependent oxidoreductase, GliT, with dithiol precursors [55].

Using the one strain many compounds (OSMAC) strategy to study the chemical diversity of *A. luteoalbus* SCSIO F457 led to one indole alkaloid, 3-(hydroxy-acetyl)-1*H*-indole (**34**, Figure 3); five cyclic dipeptides, cyclo(L-Phe-L-Pro) (**38**), cyclo(L-Tyr-L-Pro) (**39**), cyclo(L-Val-L-Pro) (**40**), cyclo(D-Ile-L-Pro) (**41**), and cyclo(D-Leu-L-Pro) (**42**); one pyranone derivative, 3-methoxy-2-methyl-4*H*-pyran-4-one (**43**); one benzo-tetrahydrofuran-lignin, paulownin (**47**); and three benzene derivatives, 1-methyoxy-4-(2-hydroxy)ethylbenzene (**48**), 2-(4-hydroxyphenyl)-ethanol (**49**), 1-phenylbutane-2,3-diol (**50**) [41].

## 4. Others

### 4.1. Cyclic Dipeptides

Three known cyclo-dipeptides; cyclo(L-Trp-L-Ser) (**35**), cyclo(L-Trp-L-Ala) (**36**), and cyclo(L-Trp-*N*-methyl-L-Ala) (**37**) (Figure 5), were isolated from the culture extract of the deep-sea sediment-derived fungus *A. luteoalbus* SCSIO F457 [37]. Although **35**–**37** showed no cytotoxic activities against cancer cell lines MCF-7, SF-268, HepG-2, and NCI-H460 with the SRB method [37], compound **35** displayed antimicrobial activity against *Escherichia coli*, *Chromobacterium violaceum* CV026, *Pseudomonas aeruginosa* PA01, *S. aureus* and *C. albicans* 00147 with the MIC values of 6.4, 3.2, 6.4, 3.2 and 6.4 mg/mL, respectively. Furthermore, **35** showed anti-quorum sensing (anti-QS) activity by inhibiting the production of violacein in *C. violaceum* CV026 with an inhibition of 67% in 0.2 mg/mL (the production inhibition of positive control azithromycin (AZM) was 80% in 0.05 mg/mL). The anti-QS activity of **35** was further confirmed by its reduction in elastase activity and biofilm formation. The reduced elastase activity in **35** was 40%, comparable with the positive control AZM, which induced a 49% inhibition. Interestingly, **35** resulted in a 59.9% reduction in biofilm formation in *P. aeruginosa* PA01 at a concentration of 0.2 mg/mL, which was better than the positive control AZM (53.9% reduction). Compound **35** or its derivatives can serve as leading compounds in the development of new antimicrobial drugs for clinical or agricultural research, playing a vital role in human health and agricultural development [56,57]. Compound **35** exhibited enzyme inhibition against *α*-glucosidase (AGS) with an IC_50_ value of 164.5 ± 15.5 µM, stronger than that of the positive control acarbose (IC_50_ = 422.3 ± 8.44 µM). In addition, **35** showed no cytotoxicity to the human normal hepatocyte (LO2) cells, suggesting its safety to be developed into hypoglycemic agent [58]. Compound **36** showed antibacterial activity against *Bacillus cereus* and *Proteus vulgaris* with MIC values of 1.56 and 3.13 µM (the MIC of positive control ciprofloxacin was 0.78 and 0.20 µM) [59]. The brine shrimp lethality of **36** was modest with an LD_50_ value of 25.5 µM (the LD_50_ of the positive control colchicine was 19.4 µM) [60]. Compound **36** exhibited 54.6 ± 0.6% cation radical (ABTS^+•^) scavenging capacity at 2 mg/mL (the positive control vitamin C displayed 79.1 ± 4.3% cation radical scavenging capacity at 0.16 mg/mL) [61]. Furthermore, **36** also showed potent anti-diatom attachment activity at the concentration of 50 µg/mL with an inhibition of 85% [62].

Further investigation of the chemical structure diversity of the fungus *A. luteoalbus* SCSIO F457, using the strategy of OSMAC, led to another five cyclic dipeptides: cyclo(L-Phe-L-Pro) (**38**), cyclo(L-Tyr-L-Pro) (**39**), cyclo(L-Val-L-Pro) (**40**), cyclo(D-Ile-L-Pro) (**41**), and cyclo(D-Leu-L-Pro) (**42**) (Figure 5) [41]. Compounds **38**–**40** could be produced by *Pseudomonas aeruginosa* to promote the growth of plant with auxin-like activity through the LasI QS system. The QS-regulated bacterial production of DKPs **38**–**40** adjusts auxin signaling and plant growth promotion, which establishes a significant function for DKPs mediating trans-kingdom signaling between prokaryote and eukaryote [63]. Compounds **38** and cyclo(L-Leu-L-Pro) showed synergistic antimicrobial activity against vancomycin-resistant enterococci (VRE) and pathogenic yeasts. The combination of **38** and cyclo(L-Leu-L-Pro) exhibited significant anti-VRE activity against *Enterococcus faecium* (K-99-38), *E. faecalis* (K-99-258), *E. faecalis* (K-99-17), *E. faecalis* (K-01-511), and *E. faecium* (K-01-312) with MIC values of 0.25–1 μg/mL. It was also effective against *E. coli*, *Micrococcus luteus*, *S. aureus*, *Cryptococcus neoformans,* and *C. albicans* with MIC values of 0.25–0.5 μg/mL. And the combination of **38** and cyclo(L-Leu-L-Pro) could reduce the mutation of strains *Salmonella typhimurium* TA98 and TA100 [64,65]. Compounds **38**–**40** displayed antifungal activities against *Ganoderma plantarum* at the concentrations of 6.8, 8.2, and 8.2 mM, respectively, and **38** also showed anti-*Candida* activity at a concentration of 7.0 mM [66]. Compounds **38** and **39** also demonstrated prominent activities against agriculturally important fungi, *Pencillium expansum*, *Rhizoctonia solani*, and *Fusarium oxysporum* with MIC values between 2 and 8 µg/mL, much higher than the commercial fungicide bavistin (MIC values was 50, 25 and 25 µg/mL, respectively) [67]. Compound **40** showed antibacterial activity against MRSA 43300 with a zone of inhibition of 15 mm at a concentration of 20 µg/disc (the inhibition zone of the positive control gentamicin was 22 mm). And **40** had low toxicity against human hepatoma HepaRG cells, meaning it could be developed into a safe antibiotic [68]. Compound **38** displayed weak cytotoxicity against HeLa, HT-29, and MCF-7 cell lines with IC_50_ values of 2.92 ± 1.55, 4.04 ± 1.15, and 6.53 ± 1.26 mM, and could induce apoptosis in HT-29 colon cancer cells [69]. The cytotoxicity of **38** in HT-29 cells could be mediated by a caspase cascade [70]. Furthermore, **38** also showed enzyme inhibition to topoisomerase I with an IC_50_ value of 13 µM, stronger than the positive control cryptotanshinone with an IC_50_ value of 17 µM [71]. Compounds **38** and **40** exhibited anti-larval activities toward barnacle *Balanus amphitrite*, with effective concentrations inhibiting 50% larval attachment (EC_50_) after 24 h of 0.28 and 0.10 mM, respectively [72,73]. And **38** and **40** also showed antioxidant activities toward OH^•^ with an inhibition of 64.9% and 54.1% at 2.5 µM, respectively [74]. Compound **42** exhibited weak cytotoxicity against ECA-109, Hela-S3, and PANC-1 cancer cells with the inhibition rates of 44%, 52%, and 55%, respectively, at 20 µM, and **42** could mildly increase the transcriptional activation of RXRα [75]. Compound **42** exhibited anti-fouling activity against cyprid larvae of the barnacle with an LC_50_ value of 3.5 μg/mL [76]. Compound **42** could obviously increase the calcium ion concentration ([Ca^2+^]_i_) in myocytes, which is heavily dependent on the extracellular Ca^2+^ influx [77]. The LPS-induced migration, adhesion, and hyperpermeability of leukocytes to a human endothelial cell monolayer and in mice could be inhibited by **42** in a dose-dependent manner, suggesting that **42** may possess the potential to be developed into therapeutic agents to treat vascular inflammatory disorders [78]. In addition, **42** was proved to suppress TGFBIp-mediated and CLP-induced septic responses, indicating that **42** could be a key candidate for therapy of the different vascular inflammatory diseases by repressing the TGFBIp signaling pathway [79].

### 4.2. Pyranone Derivatives

One pyranone derivative, 3-methoxy-2-methyl-4*H*-pyran-4-one (**43**) (Figure 5), was isolated from the culture extract of the fungus *A. luteoalbus* SCSIO F457 [41]. Compound **43** displayed no DPPH free radical scavenging or antibacterial activities [41]. In addition, **43** exhibited antibacterial activity against *S. aureus* ATCC 25923, *Enterococcus faecalis* ATCC 29212 and *E. faecium* K59–68 with MIC values of 25, 12.5, and 12.5 µg/mL, respectively [80]. The study used bioactivity tracking and molecular networking to examine the secondary metabolites of the Antarctic soil-derived fungus *A. luteoalbus* CH-6, resulting in the discovery of two new α-pyrones, acrostalapyrones A (**44**) and B (**45**), along with one previously identified analog, multiforisin G (**46**) (Figure 5) [9]. Compound **46** displayed significant immunosuppressive activity against LPS or Con A-(T-cells)-induced proliferations of mouse splenic lymphocytes (B-cells), with IC_50_ values of 1.2 and 0.9 µg/mL, respectively, which was stronger than that of positive control azathioprine (IC_50_ = 2.7 µg/mL) [81].

### 4.3. Paulownin and Benzene Derivatives

One benzo-tetrahydrofuran-lignin, paulownin (**47**), and three benzene derivatives, 1-methyoxy-4-(2-hydroxy)ethylbenzene (**48**), 2-(4-hydroxyphenyl)-ethanol (**49**), and 1-phenylbutane-2,3-diol (**50**) (Figure 5), were isolated from the deep-sea sediment-derived fungus *A. luteoalbus* SCSIO F457, using the OSMAC strategy [41]. The absolute configuration of **50** was not confirmed. Compound **48** showed antioxidant activity, and the IC_50_ of DPPH free radical scavenging of **48** was 240.05 μg/mL [41].

## 5. Conclusions

Between 1969 and 2022, researchers isolated 50 natural products from the genus *Acrostalagmus*, and 50% of these compounds are newly discovered. Between 1975 and 2011, there was a lack of research on the secondary metabolites of the genus *Acrostalagmus*, with only nine compounds isolated before 1974. However, the compounds from this genus started to attract the attention of researchers after 2012. Interestingly, all the compounds isolated between 2012 and 2022 are derived from the marine *Acrostalagmus*, and they comprise 82% of the natural products discovered from this genus (Table 1). Thess findings highlight the ocean as a vast resource treasury and suggest that the marine-derived genus *Acrostalagmus* possesses the ability to produce abundant secondary metabolites.

The compounds isolated from the genus *Acrostalagmus* exhibit diverse structures, including terpenoids, alkaloids, peptides, pyranones, benzene derivatives, and paulownin. Among these compounds, alkaloids are of particular importance, comprising 56% of the natural products derived from this genus (Figure 6). Furthermore, within the alkaloid class, 61% belong to the epipolythiodioxopiperazine family. This substantial proportion highlights the significance of epipolythiodioxopiperazine as a key characteristic structure within the genus *Acrostalagmus*.

The genus *Acrostalagmus* has the potential to produce a variety of secondary metabolites with diverse bioactivities, including plant growth regulation, enzyme, Hsp90, and biofilm inhibitions, cytotoxic, antimicrobial, nematicidal, anti-inflammatory, immunosuppressive, antifouling, anti-QS, brine shrimp lethal, and antioxidant activities (Figure 7). Research indicates that 72% of the natural products obtained from *Acrostalagmus* exhibit bioactive activities, with compounds **1**, **10**, **12**, **13**, **15**, **32**, **35**, **36**, **38**, **40**, and **42** displaying more than three types of activity, and 50% of the bioactive compounds exhibiting prominent activities comparable or stronger than their positive controls, which further demonstrates the potential ability of this genus to produce bioactive natural products (Figure 7). 

The bioactive compounds isolated from the genus *Acrostalagmus* mainly focus on cytotoxic (19%), enzyme inhibitory (17%), and antimicrobial (29%, with antibacterial (17%) and antifungal (12%) activities) activities (Figure 8), indicating considerable potential for the development of new anticancer compounds, enzyme inhibitors, and antibiotics from *Acrostalagmus*.

According to research, 72% of natural products derived from *Acrostalagmus* display bioactivities, with 50% of the bioactive compounds exhibiting more significant or comparable activities than their positive controls (Table 2, Table 3 and Table 4). Most of the compounds with remarkable activities (67%) belong to the family of epipolythiodioxopiperazine, confirming the potential of this structure as a precursor for the development of novel drugs. Eighty-nine percent of potent active compounds are isolated from marine derived fungi, further demonstrating the development potential of marine fungi.

The stronger cytotoxic activities of compounds **3** and **10**–**17** compared to their positive control (Figure 7, Table 2) support their potential as new anticancer drugs. Compounds **9**, **12**, **31,** and **32** exhibit more significant antibacterial activities than their positive controls, and **16** and **30** show comparable antibacterial activities compared to their positive control (Figure 7, Table 3), meaning they could be valuable starting points for the development of new antibiotics. Notably, compounds **9**, **12,** and **32** demonstrate stronger antibacterial activities against MRSA than their positive controls (Figure 7, Table 3), addressing the challenge of bacterial drug resistance. The combination of **38** and cyclo(L-Leu-L-Pro) exhibited obvious synergistic effect, with significant antimicrobial activity against VRE and pathogenic yeasts, which supports their potential use as synergistic antibiotics. Compounds **38** and **39** demonstrate prominent activities against agriculturally important fungi, much higher than the commercial fungicide bavistin, declaring the potential of **38** and **39** to be applied in agricultural fungicide (Figure 7, Table 3). 

Compound **1** exhibits greater inhibition of germination and growth development at a concentration of 10^−4^ M compared to the commercial herbicide LOGRAN^®^. This indicates the potential for developing compound **1** as a new herbicide (Figure 7, Table 4). Compound **15** displays more potent inhibition of mushroom tyrosinase compared to the positive control kojic acid (Figure 7, Table 4), which demonstrates that **1** could be employed in various fields such as whitening and health care, treatment of pigmented skin diseases, pest control, and food preservation. Compound **35** exhibits stronger inhibition of the biofilm formation in *P. aeruginosa* PA01 than the positive control AZM, indicating **35** can serve as leading compound in developing new antimicrobial drugs for clinical or agricultural research. Compound **35** also shows more significant enzyme inhibition against *α*-glucosidase (AGS) than the positive control acarbose. In addition, **35** shows no cytotoxicity to the human normal hepatocyte (LO2) cells, suggesting the safety of **35** to be developed into hypoglycemic agent (Figure 7, Table 4). Compound **38** displays potent enzyme inhibition to topoisomerase I, stronger than the positive control cryptotanshinone, suggesting it can be developed into new antitumor drugs (Figure 7, Table 4). Compound **46** displays significant immunosuppressive activity, stronger than that of positive control azathioprine, which has the potential to be developed into immunomodulatory drugs (Figure 7, Table 4). These results further suggest that the genus *Acrostalagmus* holds promise as a source of bioactive compounds.

In this review, we comprehensively summarized the chemical structure types, biosynthesis, bioactivity, sources, and distribution of the secondary metabolites isolated from *Acrostalagmus* in the period between 1969 and 2022. The literature survey indicates that *Acrostalagmus*, especially marine derived *Acrostalagmus*, has great potential to produce abundant and diverse new bioactive natural products, and the family of epipolythiodioxopiperazine, with its significant bioactivities, could be one of the characteristic compound groups of the genus *Acrostalagmus*.

## Figures and Tables

**Figure 1 biomolecules-13-01191-f001:**
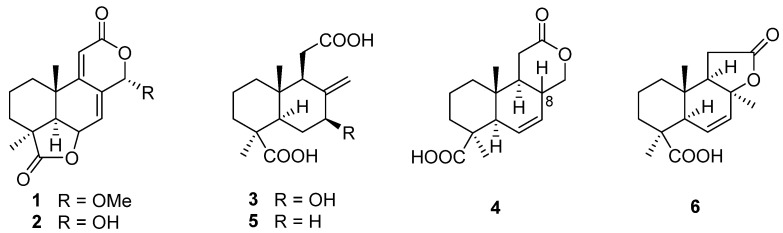
Chemical structures of compounds **1**–**6** [19,20,21,22].

**Figure 2 biomolecules-13-01191-f002:**
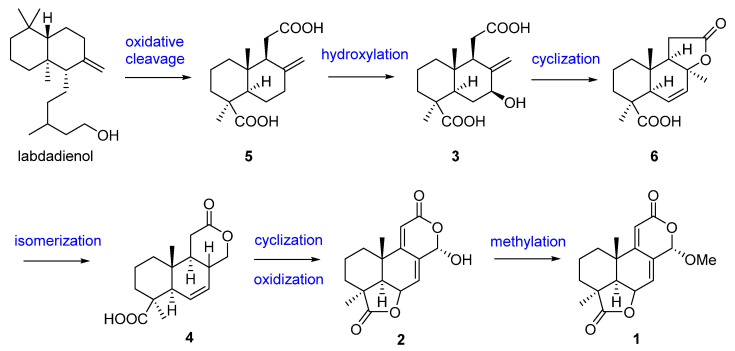
Presumed biosynthesis pathway of compounds **1**–**6** [22].

**Figure 3 biomolecules-13-01191-f003:**
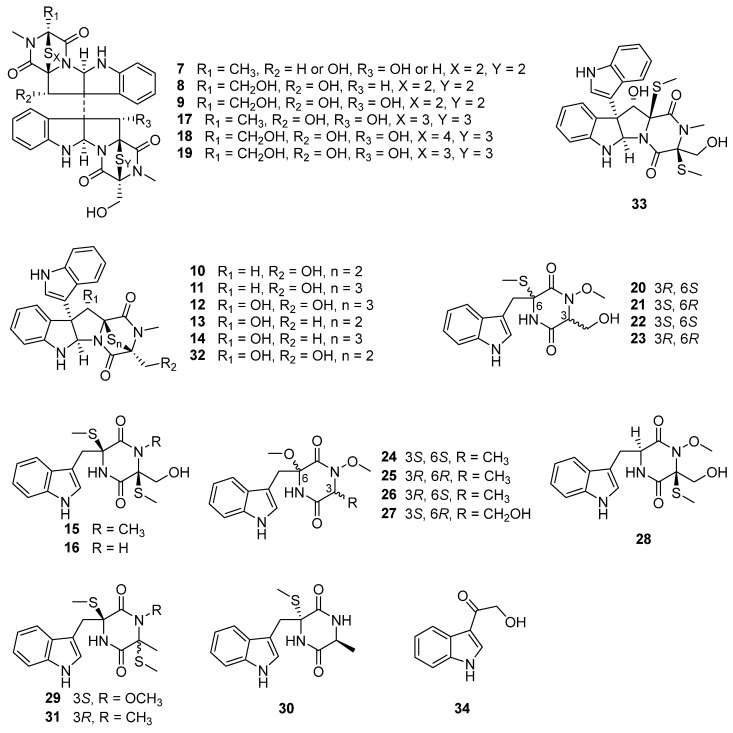
Chemical structures of compounds **7**–**34** [34,37,38,39,40,41].

**Figure 4 biomolecules-13-01191-f004:**
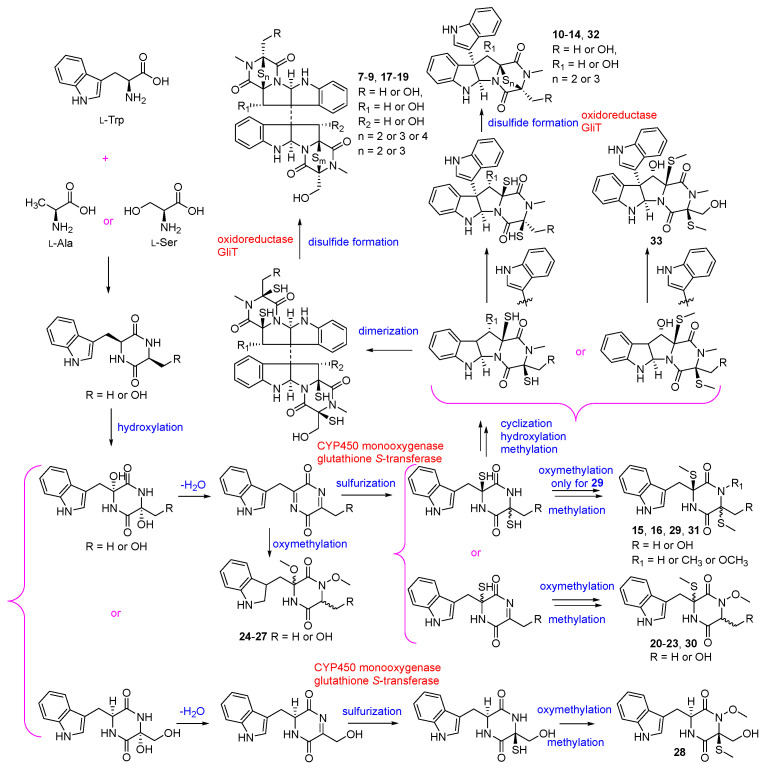
Proposed biosynthesis of compounds **7**–**33** [40,48,51,52,53,54,55].

**Figure 5 biomolecules-13-01191-f005:**
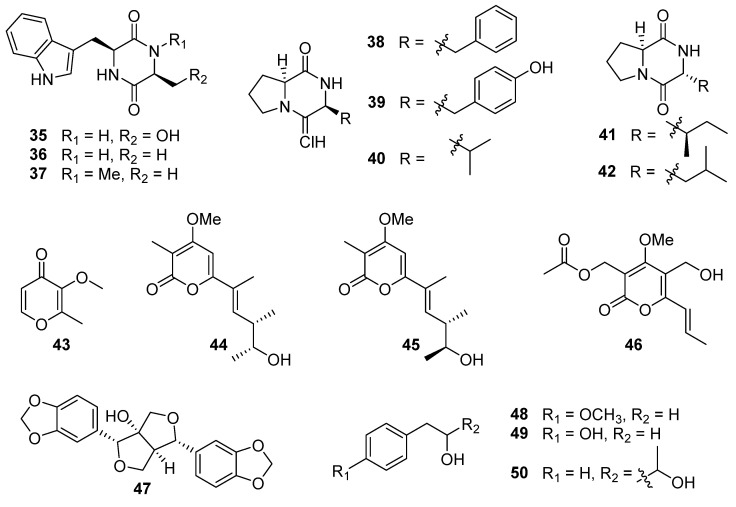
Chemical structures of compounds **35**–**50** [9,37,55].

**Figure 6 biomolecules-13-01191-f006:**
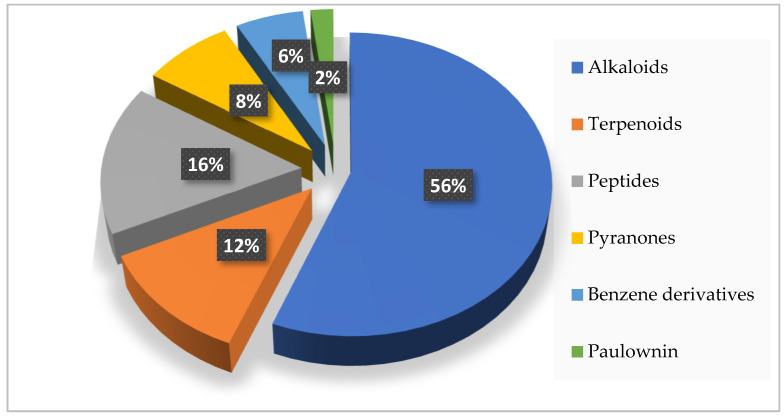
Structural types of compounds isolated from *Acrostalagmus* during 1969–2022.

**Figure 7 biomolecules-13-01191-f007:**
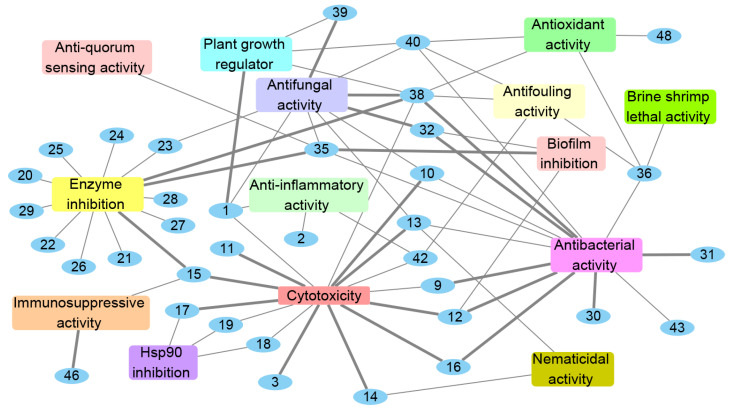
Bioactivities of natural products isolated from *Acrostalagmus* during 1969–2022. The bold edges mean compounds with strong activities.

**Figure 8 biomolecules-13-01191-f008:**
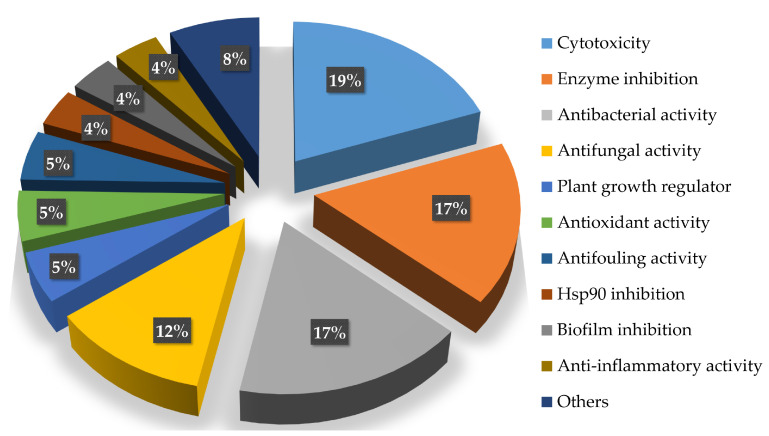
Percentage of *Acrostalagmus* isolated compounds with different bioactivities during 1969–2022.

**Table 1 biomolecules-13-01191-t001:** Compounds isolated from *Acrostalagmus* during 1969–2022.

Types	Compounds	Sources	Distribution	Years	Refs.
Terpenoids	**1**, **2**	*Acrostalagmus* sp. NRRL-3481		1969	[19]
	**3**			1971	[21]
	**4**–**6**			1974	[22]
Alkaloids	**7**–**9**	*A. cinnabarinus* var. *melinacidinus*		1972	[33]
	**10**–**16**	Deep-sea sediment-derived fungus *A. luteoalbus* SCSIO F457 (GenBank No. MN860118)	South China Sea	2012	[37]
	**17**–**19**	Soil derived fungus *A. luteoalbus* HDN13-530 (GenBank No. KP969081)	Liaodong Bay, China	2017	[38]
	**20**–**25**	Marine green alga *Codium fragile* derived endophytic fungus *A. luteoalbus* TK-43 (GenBank No. MH836621)	Sinop, Turkey	2019	[39]
	**26**–**34**			2021	[40]
Peptides	**35**–**37**	Deep-sea sediment-derived fungus *A. luteoalbus* SCSIO F457 (GenBank No. MN860118)	South China Sea	2012	[37]
	**38**–**42**			2020	[41]
Pyranone derivatives	**43**	Deep-sea sediment-derived fungus *A. luteoalbus* SCSIO F457 (GenBank No. MN860118)	South China Sea	2020	[41]
	**44**–**46**	Antarctic soil derived fungus *A. luteoalbus* CH-6 (Genbank No. MT367202.1)	Fields Peninsula, Antarctica	2022	[9]
Paulownin	**47**	Deep-sea sediment-derived fungus *A. luteoalbus* SCSIO F457 (GenBank No. MN860118)	South China Sea	2020	[41]
Benzene derivatives	**48**–**50**	Deep-sea sediment-derived fungus *A. luteoalbus* SCSIO F457 (GenBank No. MN860118)	South China Sea	2020	[41]

**Table 2 biomolecules-13-01191-t002:** Cytotoxicity of compounds isolated from *Acrostalagmus* during 1969–2022.

Cell Lines	Compounds	Values (IC_50_)	Values of Positive Controls (IC_50_)	Pros and Cons
P388	**1** (μg/mL)	4.1		Pros: Strong and broad spectrum cytotoxicity [23].
BXPC-3		0.36		
MCF-7		0.33		
SF268		0.24		
NCI-H460		0.24		
KM20L		0.21		
DU-145		0.14		
HL-60	**3** (μM)	**0.60**	0.71	Pro: Strong cytotoxicity with the same level as the positive control [25].
P388	**9/12** (μM)	0.05/0.25		Pro: Potent cytotoxicity against murine P388 leukemia cells [35,43].
SF-268	**10/11/12**/ **13/14** (μM)	**0.46 ± 0.05/0.59 ± 0.03/1.04 ± 0.03**/ **0.73 ± 0.05/2.49 ± 0.07**	4.76 ± 0.27	Pros: Compounds **10**–**14** exhibited potent cytotoxicity, and **10** and **11** showed stronger cytotoxicity against all four tested cancer cell lines than that of the positive control cisplatin [37].
MCF-7		**0.23 ± 0.03/0.25 ± 0.00/0.91 ± 0.03**/ **0.23 ± 0.03/0.65 ± 0.07**	3.99 ± 0.13	
NCI-H460		**1.15 ± 0.03/1.31 ± 0.12**/5.60 ± 0.58/ 6.57 ± 0.81/17.78 ± 0.27	2.91 ± 0.18	
HepG-2		**0.91 ± 0.03/1.29 ± 0.16**/3.52 ± 0.74/ **0.53 ± 0.04/2.03 ± 0.07**	2.45 ± 0.07	
A549	**10/11** (μM)	2.33 ± 0.59/0.91 ± 0.29		Pro: Prominent cytotoxic activities [42].
HeLa		1.00 ± 0.24/0.52 ± 0.15		
HCT116		1.22 ± 1.02/0.58 ± 0.38		
L5178Y	**15/16** (μM)	**0.26/0.82**	4.3	Pro: Potent cytotoxic activities against murine lymphoma L5178Y cell line, which are more potent than that of the positive control kahalalide F [46].
A549	**17/18/19** (μM)	0.4/1.9/0.7	0.2	Pro: Extensive cytotoxicity, **17** showed stronger activity to H1975 than that of positive drug doxorubicin hydrochloride [38]
HCT116		0.4/2.1/0.3	0.2	
K562		0.4/1.9/1	0.2	
H1975		**0.2**/3.6/0.8	0.8	
HL-60		1.9/1.9/1.5	0.02	
HCT-8	**19** (μM)	0.49 ± 0.09		Pro: Significant cytotoxicity against a panel of cancer cell lines [48]
Bel-7402		0.38 ± 0.03		
BGC-823		0.70 ± 0.04		
A2780		0.58 ± 0.03		
HeLa	**38** (mM)	2.92 ± 1.55		Con: Weak activity [69].
HT-29		4.04 ± 1.15		
MCF-7		6.53 ± 1.26		
ECA-109	**42** (inhibition rate at 20 µM)	44%		Con: Weak activity [75].
Hela-S3		52%		
PANC-1		55%		

The bold cytotoxic values are stronger than their positive controls.

**Table 3 biomolecules-13-01191-t003:** Antimicrobial activities of compounds isolated from *Acrostalagmus* during 1969–2022.

Strains	Compounds	Values (MIC)	Values of Positive Controls (MIC)	Pros and Cons
*Cryptococcus neoformans* ATCC 90112	**1** (μg/mL)	2		Pro: Strong activity against fungus *C. neoformans* caused infection in human [23,24].
*Candida albicans* ATCC 90028		8		
*Pseudogymnoascus destructans* ATCC MYA 4855		15		
methicillin-resistant *Staphylococcus aureus* (MRSA)	**9** (μg/mL)	**0.7**	1.4	Pros: Strong antibacterial activity to MRSA, the activity was double of the positive control [36].
vancomycin-resistant *Enterococcus faecium* (VRE)		22	2.4	
*S. aureus* ATCC29213	**12/32** (μM)	3.8 ± 0.40/5.8 ± 0.45	0.362 ± 0.09	Pros: Broad-spectrum antimicrobial activity; Strong activity against MRSA compared with positive control. Con: Moderate or weak antimicrobial activity to some of the test strains [50].
MRSA		**8.4 ± 1.01/5.6 ± 0.99**	9.33 ± 2.6	
*Bacillus cereus* IIIM25		9.2 ± 0.77/9.9 ± 0.81	0.12 ± 0.009	
*Klebsiella pneumoniae* ATCC75388		19.1 ± 1.1/4.5 ± 0.77	0.015 ± 0.0006	
*Bacillus thuringiensis* MTCC 809		14.8 ± 0.28/19 ± 0.84	0.003 ± 0.001	
*Yersinia enterocolitica* MTCC840		38 ± 1.7/65.3 ± 1.6	3.5 ± 0.202	
*Erwinia herbicola* MTCC3609		15.4 ± 2.7/14.2 ± 1.4	0.006 ± 0.0009	
*Shigella dysenteriae* NCTC 11311		82.3 ± 1.3/–	0.006 ± 0.0003	
*Lactococcus lactis* MTCC440		28.7 ± 1.7/39.4 ± 1.1	0.006 ± 0.001	
*S. epidermidis* MTCC35		22.6 ± 2.2/23.4 ± 1.5	0.06 ± 0.006	
*Alcaligenes faecalis* MTCC126		–/–	1.2 ± 0.06	
*S. warneri* MTCC4436		5.05 ± 0.4/7.5 ± 0.4	2.4 ± 0.105	
*Pseudomonas fluorescens* MTCC103		18.4 ± 0.3/26.1 ± 2.7	0.151 ± 0.051	
*Xanthobacter flavus* MTCC 132		98.3 ± 1.1/–	2.3 ± 0.021	
*S. pyogenes* MTCC442		1.8 ± 0.2/3.1 ± 0.15	0.015 ± 0.0006	
*Shigella boydii* NCTC9357		31.5 ± 1.2/26.7 ± 0.9	1.12 ± 0.063	
*Clostridium pasteurianum* MTCC116		92.3 ± 0.4/54.0 ± 0.5	0.015 ± 0.003	
*Salmonella typhimurium* MTCC98		–/86.2 ± 1.9	0.015 ± 0.003	
*C. albicans* MTCC4748		–/35.8 ± 1.4	1.5 ± 0.022	
*C. albicans*	**10/13/32** (μM)	12.5/25/**6.25**	6.25	Pro: Compound **32** showed broad-spectrum antimicrobial activity. Con: Weak activity [9].
*Aeromonas salmonicida*		12.5/50/**3.125**	6.25	
*Photobacterium halotolerans*		-/-/25	0.195	
*Pseudomonas fulva*		-/-/25	1.56	
*S. aureus*		-/-/25	3.125	
*Escherichia coli*	**16/30/32** (μM)	-/-/**8**	12	Pros: Compound **32** showed broad-spectrum antimicrobial activity, and the activity is significant and comparable to that of the positive control; compounds **16** and **30** displayed specific remarkable antibacterial activities toward *Ed. ictaluri* [40].
*Edwardsiella tarda*		-/-/**2**	2	
*Ed. ictaluri*		**5/3/2**	2	
*Aeromonas hydrophila*		-/-/**4**	3	
*Micrococcus luteus*		-/-/33	3	
*Pseudomonas aeruginosa*		-/-/**8**	6	
*Vibrio alginolyticus*		-/-/8	2	
*V. anguillarum*		-/-/**2**	3	
*V. harveyi*		-/-/**4**	3	
*V. parahemolyticus*		-/-/**2**	12	
*V. vulnificus*		-/-/33	3	
*Fusarium solani*	**23** (μg/mL)	32		Pro: **23** exhibited specific antifungal activity toward *F. solani* [39].
*Veillonella parvula*	**31** (μg/mL)	0.25	0.12	Pro: **31** exhibited strong antibacterial activity, comparable or even more significant than that of positive control [49].
*Actinomyces israelii*		32	8	
*Streptococcus* sp.		**0.12**	0.25	
*Bacteroides vulgatus*		**0.12**	0.5	
*Peptostreptococcus* sp.		**0.12**	0.5	
*E. coli*	**35** (mg/mL)	6.4		Con: Weak activity [56,57].
*Chromobacterium violaceum* CV026		3.2		
*Pseudomonas aeruginosa* PA01		6.4		
*S. aureus*		3.2		
*C. albicans* 00147		6.4		
*B.* *cereus*	**36** (μM)	1.56	0.78	Con: Medium activity [59].
*Proteus vulgaris*		3.13	0.20	
*Enterococcus faecium* (K-99-38)	**38** and cyclo(L-Leu-L-Pro)/**38** (μg/mL)	**1/64**	64	Pro: Combination of **38** and cyclo(L-Leu-L-Pro) displayed prominent antimicrobial activity, much stronger than those of positive controls [64,65].
*E. faecalis* (K-99-17)		**0.5/16**	128	
*E. faecalis* (K-99-258)		**0.25/32**	>256	
*E. faecalis* (K-01-312)		**2/16**	128	
*E. faecium* (K-01-511)		**0.5/32**	128	
*E. col*		**0.5**/64	32	
*B. subtilis*		**1**/128	64	
*Micrococcus luteus*		**0.25**/64	32	
*S. faecalis*		**2**/>256	64	
*P. aeruginosa*		**1**/64	12.5	
*S. aureus*		**0.5**/256	25	
Penicillin resistant *S. aureus*		**4**/256	64	
*C. albicans*		**0.25**/64	32	
*C. glabrata*		**4**/256	16	
*C* *. tropicalis*		**0.5**/32	128	
Amphotericin B resistant *C. tropicalis*		**0.5**/64	16	
*Cryptococcus neoformans*		**0.25**/32	16	
Amphotericin B resistant *C. neoformans*		**2**/>256	32	
*Ganoderma plantarum*	**38/39/40** (mM)	6.8/8.2/8.2		Con: Weak activity [66].
*Candida* sp.		7.0		
*B. subtilis* MTCC2756	**38/39** (µg/mL)	16/64	5	Pro: Demonstrated prominent activities against agriculturally important fungi, much higher than the commercial fungicide bavistin [67]
*S. aureus* MTCC902		16/32	5	
*E. coli* MTCC2622		8/32	5	
*P. aeruginosa* MTCC2642		32/-	10	
*Aspergillus flavus* MTCC183		128/32	100	
*C. albicans* MTCC277		64/32	50	
*Fusarium oxysporum* MTCC284		**4/8**	25	
*Rhizoctonia solani* MTCC4634		**4/8**	25	
*Pencillium expansum* MTCC2006		**2/4**	50	
MRSA 43300 (inhibition zone)	**40** (mm)	15	22	Con: Medium activity [68].
*S. aureus* ATCC 25923	**43** (µg/mL)	25		Con: Medium activity [80].
*Enterococcus faecalis* ATCC 29212		12.5		
*E. faecium* K59–68		12.5		

The bold antimicrobial values are stronger or comparable than their positive controls.

**Table 4 biomolecules-13-01191-t004:** Other bioactivities of compounds isolated from *Acrostalagmus* during 1969–2022.

Bioactivities	Cells/Stains/Enzyme	Compounds	Values	Values of Positive Controls	Pros and Cons
Plant growth regulator, inhibition of the germination and growth development at 10^−4^ M (%)	*Avena coleoptile*	**1**			Pro: Significant inhibitory activity, and more active than the commercial herbicide LOGRAN^®^ [26,27].
	*Allium cepa*		**>80%**	65%	
	*Hordeum vulgare*		**>80%**	<60%	
	*Lactuca sativa*		**>80%**	<60%	
Plant growth regulator	Auxin signaling and plant growth promotion	**38** **–** **40**			Pro: Established a significant function for DKPs mediating transkingdom signaling between prokaryote and eukaryote [63].
Anti-inflammatory activity (IC_50_, μM)	IL-1β	**1/2**	0.049/69		Pro: Compound **1** showed potent inhibitory activity to the production of IL-1β. Con: Compound **2** showed weak activity [28,29,30].
	TNF-α		3.0/11		
	Leucine uptake		11/120		
	Inhibition the LPS-induced migration, adhesion, and hyperpermeability of leukocytes	**42**			Pro: Potential candidate for therapy of the different vascular inflammatory diseases [78,79]
	Suppress TGFBIp-mediated and CLP-induced septic responses				
Nematicidal activity (ED_50_, µg/mL)	*Caenorhabditis elegans*	**13/14**	200/200		Con: Weak activities [44].
	*Panagrellus redivivus*		250/250		
Biofilm inhibition at MIC values (%)	*S.aureus*	**12/32**	70.3%/ 68.8%		Pros: Strong activities [50], **35** displayed stronger biofilm inhibition than that of positive control azithromycin [56,57].
	*S. pyogenes*		60.75%/ 86.4%		
	*Pseudomonas aeruginosa* PA01	**35** (1/32 MIC)	**59.9%**	53.9%	
Immunosuppressive activity, IC_50_ value on Con A-(T-cells)-induced or LPS-induced proliferations of mouse splenic lymphocytes (µg/mL)	Con A-(T-cells)-induced	**15/46**	24/**0.9**	2.7	Pro: Compound **46** showed significant immunosuppressive activity and stronger than that of positive control azathioprine [81]. Con: Weak activity of **15** [45].
	LPS-induced	**46**	**1.2**	2.7	
Enzyme inhibition (IC_50_, µM)	Mushroom tyrosinase	**15**	**31.7 ± 0.2**	40.4 ± 0.1	Pro: Stronger than the inhibition of the positive control kojic acid [47].
	AChE	**20** and **21**	9.5	0.14	Pro: Compounds **35** and **38** with stronger enzyme inhibition than their positive control acarbose [58,71]. Con: Medium or weak activity [39,40].
		**20/21**	2.3/13.8		
		**22** and **23**	60.7		
		**22/23**	78.8/49.2		
		**24** and **25**	130.5		
		**24/25**	160.6/ 121.7		
		**26/27**/ **28/29**	18.9/32/ 8.4/32		
	*α*-Glucosidase	**35**	164.5 ± 15.5	422.3 ± 8.44	
	Topoisomerase I	**38**	**13**	17	
Hsp90 inhibition at the concentration of 0.5 μM	H1975 cells	**17/18/19**			Reduce the expressions of Akt, EGFR, and the active forms of Akt, EGFR, Erk, and Stat3 (Hsp90 client oncoproteins) [38].
Anti-quorum sensing activity (0.2 mg/mL)	Inhibiting the production of violacein in *Chromobacterium violaceum* CV026	**35**	67%	80% in 0.05 mg/mL	Pro: Strong activity [56,57].
	Reduction in elastase activity		40%	49% in 0.05 mg/mL	
Brine shrimp lethality (LD_50_, μM)		**36**	25.5	19.4	Con: Medium activity [60].
Antifouling activity	Anti-diatom attachment activity	**36** (50 µg/mL)	85%		Pro: Strong activities of **36** and **42** [62,76]. Con: Weak activities of **38** and **40** [72,73].
	*Balanus amphitrite* (EC_50_)	**38/40** (mM)	0.28/0.10		
	Cyprid larvae of the barnacle (LC_50_)	**42** (μg/mL)	3.5		
Antioxidant activity, DPPH free radical scavenging	ABTS^+•^ scavenging capacity at 2 mg/mL	**36**	54.6 ± 0.6%	79.1 ± 4.3% at 0.16 mg/mL	Con: Medium or weak activities [41,61,74].
	OH^•^ inhibition at 2.5 μM	**38/40**	64.9%/54.1%		
	IC_50_ (μg/mL)	**48**	240.05	16.87	

The bold bioactive values are stronger than their positive controls.

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
