# Peer review of "Genus Acrostalagmus: A Prolific Producer of Natural Products"

_biomolecules, 2023, doi:10.3390/biom13081191_

Round 1

Reviewer 1 Report

The authors of this review article have summarized extremely important and scientifically interesting information about biologically active compounds produced by the genus Acrostalagmus in the family Plectosphaerellaceae over an impressive period of time. The authors reviewed and presented in sufficient detail the chemical structure types, sources, distribution, biological activities, and biological synthesis of the compounds isolated from Acrostalagmus.  This information can be useful for researchers involved in both fundamental and applied sciences. Currently, the direction of searching for biologically active substances with multitarget action is extremely relevant. And in this case, the genus of ascomycete fungi Acrostalagmus, especially marine derived Acrostalagmus and other marine derived fungi have  great potential for further research for the development of novel drugs.

A few points:

In Sections 4.1 and 4.2, the authors presented only the chemical structure of cyclic dipeptides and pyranone derivatives isolated from a culture extract of the fungus A. luteoalbus SCSIO F457. However, there is no information on the biological activity and prospects of these compounds for both drug development and other biological products.

Author Response

Dear reviewer:

Thanks very much for your effort for our paper entitled "Genus Acrostalagmus: A Prolific Producer of Natural Products". We have revised our manuscript carefully according to your comments. The changes in our new version of manuscript were highlighted in red.

Question

In Sections 4.1 and 4.2, the authors presented only the chemical structure of cyclic dipeptides and pyranone derivatives isolated from a culture extract of the fungus A. luteoalbus SCSIO F457. However, there is no information on the biological activity and prospects of these compounds for both drug development and other biological products.

Answer

Thanks very much for your kind comment. The information on the biological activity and prospects of these compounds for both drug development and other biological products of the cyclic dipeptides and pyranone derivatives isolated from a culture extract of the fungus A. luteoalbus SCSIO F457 has been supplied in Sections 4.1 and 4.2.

Reviewer 2 Report

The manuscript “Genus Acrostalagmus: A Prolific Producer of Natural Products” [biomolecules-2507510-peer-review-v1] written by Ting Shi, Han Wang, Yan-Jing Li, Yi-Fei Wang, Qun Pan, Bo Wang and Er-Lei Shang contains a review dealing with all natural products isolated from the genus Acrostalagmus since the first natural product was isolated in 1969. The authors focus on the structures of the isolated natural products and group then to different classes of natural products. Earlier reported suggestions of the biosynthesis are summarized. Furthermore, the tested bioactivities of these natural products are reviewed to certain detail and discussed with respect to possible pharmaceutical applications.

The review is clear organized and pleasant to read. The selected and presented results and data are comprehensively listed and the quoted literature seem to completely cover the selected research theme, including data from rather new publications of the reviewed research field. It is therefore a comprehensive review that summarizes and illuminates different aspects of bioactive natural products from Acrostalagmus.

Therefore, the manuscript is of interest in the fields of Mycology, Marine Chemistry, Pharmaceutical Chemistry, and Natural Product Chemistry in general. It has a strong relation to research in biomolecular research. Hence it is worth publishing in “Biomolecules”. However, the reviewer has a few general and minor comments, which should be taken into account by the authors in a revision prior to acceptance of the manuscript:

General comments:

a) In Figures 1, 3, 4 and 5, the authors have not indicated all stereo centers using Natta Projection. Here it would be useful to point out that this was not defined in the original publications. (Or correct this if it has been forgotten. This is certainly the case with the amino acids in Figures 4 and 5.) It would also be appropriate to at least critically mention this fact in context of the possible biosynthesis and, if necessary, to point out probable absolute and relative configurations.

b) Figure 4: Tryptamine lacks a double bond in the heterocyclic ring. This error also runs through many subsequent follow up products. The authors are urged to correct this. // The step to compounds 15, 16, 29, 30 does not always contain an “oxymethylation”.

Minor comments:

c) In Figures 7 and 8 as well as Table 4 (and the associated texts) it would make sense to point out more intensively that the so far described bioactivities (and not all possible) are summarized. Certainly not all bioactivities have been exhaustively tested and described in every publication.

d) Line 64: What is a “normal diterpene”? Just “diterpene” would be sufficient to illustrate the C20 carbon skeleton. Or the authors should directly focus on “bicyclophytane” as group of diterpenes.

Author Response

Dear reviewer:

Thanks very much for your effort for our paper entitled "Genus Acrostalagmus: A Prolific Producer of Natural Products". We have revised our manuscript carefully according to your comments. The changes in our new version of manuscript were highlighted in red.

Question 1

In Figures 1, 3, 4 and 5, the authors have not indicated all stereo centers using Natta Projection. Here it would be useful to point out that this was not defined in the original publications. (Or correct this if it has been forgotten. This is certainly the case with the amino acids in Figures 4 and 5.) It would also be appropriate to at least critically mention this fact in context of the possible biosynthesis and, if necessary, to point out probable absolute and relative configurations.

Answer 1

Thanks very much for your kind comment. In Figures 1–5, the stereo centers of the compounds have been supplied. The absolute configuration of 4 at the location of C-8 was deduced to be 8R according to the supposed biosynthesis pathway from compound 6 to 4. And The absolute configuration of 50 was not confirmed.

Question 2

Figure 4: Tryptamine lacks a double bond in the heterocyclic ring. This error also runs through many subsequent follow up products. The authors are urged to correct this. // The step to compounds 15, 16, 29, 31 does not always contain an “oxymethylation”.

Answer 2

Thanks very much for your kind comment. The wrong structures in Figure 4 have been revised, and only compound 29 contain the reaction of oxymethylation, which has been supplied in Figure 4.

Question 3

In Figures 7 and 8 as well as Table 4 (and the associated texts) it would make sense to point out more intensively that the so far described bioactivities (and not all possible) are summarized. Certainly not all bioactivities have been exhaustively tested and described in every publication.

Answer 3

Thanks very much for your kind suggestion. In Figure 7 and Tables 2–4, the significant bioactive activities have been bold to point out. The associated texts have been supplied in lines 363, 364, 385–397 and 402–420. In Figure 8, some activities with a small number of compounds were put into other categories.

Question 4

Line 64: What is a “normal diterpene”? Just “diterpene” would be sufficient to illustrate the C20 carbon skeleton. Or the authors should directly focus on “bicyclophytane” as group of diterpenes.

Answer 4

Thanks very much for your kind suggestion. The “normal” has been delete in line 64.